# Magnetic domains and domain wall pinning in atomically thin CrBr$_3$ revealed by nanoscale imaging

Qi-Chao Sun [1,8✉], Tiancheng Song [2,8], Eric Anderson [2], Andreas Brunner[1], Johannes Förster[3],
Tetyana Shalomayeva[1], Takashi Taniguchi [4], Kenji Watanabe [4], Joachim Gräfe [3], Rainer Stöhr[1,5✉],
Xiaodong Xu [2,6] & Jörg Wrachtrup[1,7]

The emergence of atomically thin van der Waals magnets provides a new platform for the studies of two-dimensional magnetism and its applications. However, the widely used measurement methods in recent studies cannot provide quantitative information of the magnetization nor achieve nanoscale spatial resolution. These capabilities are essential to explore the rich properties of magnetic domains and spin textures. Here, we employ cryogenic scanning magnetometry using a single-electron spin of a nitrogen-vacancy center in a diamond probe to unambiguously prove the existence of magnetic domains and study their dynamics in atomically thin CrBr$_3$. By controlling the magnetic domain evolution as a function of magnetic field, we find that the pinning effect is a dominant coercivity mechanism and determine the magnetization of a CrBr$_3$ bilayer to be about 26 Bohr magnetons per square nanometer. The high spatial resolution of this technique enables imaging of magnetic domains and allows to locate the sites of defects that pin the domain walls and nucleate the reverse domains. Our work highlights scanning nitrogen-vacancy center magnetometry as a quantitative probe to explore nanoscale features in two-dimensional magnets.

[1] 3. Physikalisches Institut, University of Stuttgart, Stuttgart, Germany. [2] Department of Physics, University of Washington, Seattle, WA, USA. [3] Max Planck Institute for Intelligent Systems, Stuttgart, Germany. [4] National Institute for Materials Science, Tsukuba, Ibaraki, Japan. [5] Center for Applied Quantum Technology, University of Stuttgart, Stuttgart, Germany. [6] Department of Materials Science and Engineering, University of Washington, Seattle, WA, USA. [7] Max Planck Institute for Solid State Research, Stuttgart, Germany. [8] These authors contributed equally: Qi-Chao Sun, Tiancheng Song. ✉email: q.sun@pi3.uni-stuttgart.de; rainer.stoehr@pi3.uni-stuttgart.de

The discovery of atomically thin van der Walls (vdW) magnetic materials enables fundamental studies of magnetism in various spin systems in the two-dimensional (2D) limit[1,2]. Advantages, such as easy fabrication and a wide variety of control mechanisms[3–9], make the vdW magnets and their heterostructures promising candidates for next-generation spintronic devices. This combination of fundamental and technological interest has motivated both the search for new room-temperature vdW magnets and the investigation of the mechanisms determining magnetic properties of already discovered materials. The vdW magnets have been intensively studied at micrometer scale by several probe techniques, such as magneto-optical Kerr effect microscopy[3,4], magnetic circular dichroism microscopy[6,10], anomalous hall effect[11,12], and electron tunneling[8,13,14]. Despite many significant results, these methods have limited spatial resolution due to the laser diffraction limit and electrode size. The nanoscale features of the atomically thin vdW magnets, such as magnetic domains[15–17] and topological spin textures[18–21], are largely unexplored. For example, magnetic domains in layered $CrBr_3$ have been predicted from its anomalous hysteresis loop in magneto-photoluminescence and micromagnetometry measurements[15,16], but the magnetic domain structure and its evolution has not been detected in real space. As a ferromagnetic insulator, $CrBr_3$ provides a unique spin system to study the ferromagnetism and spin fluctuation in the 2D limit[10,15,16], and is a building block of vdW heterostructures[22–25]. Quantitatively studying the magnetic properties at the nanoscale would allow a better understanding of the material and benefit device design.

Many high-spatial-resolution magnetic imaging techniques, such as magnetic force microscopy and Lorentz transmission electron microscopy, have been successfully used to study magnetic thin film materials. However, recent works show that it remains challenging to probe atomically thin vdW magnets with them due to the weak signal level[17,20]. Scanning superconducting quantum interference device can achieve very high magnetic field sensitivity even with a probe diameter of ~50 nm (refs. [26,27]). The method works well at temperatures below a few Kelvin, but the work temperature range is limited by the low-temperature superconducting material of the probe. In contrast, the negatively charged nitrogen-vacancy (NV) center has been demonstrated as a high sensitivity magnetometer with operational temperatures from below one to several hundreds of Kelvin[28], which is suitable to probe most of the discovered vdW magnets. Scanning magnetometry combining atomic force microscopy and NV center magnetometers allows for quantitative nanoscale imaging of magnetic fields, and has been well established in room-temperature measurements[29–34]. Recently, its cryogenic implementation has been demonstrated[35,36] and has already been employed to image the magnetization in layered $CrI_3$ (ref. [37]). In these works, the magnetic field is measured using the continuous wave (cw) optically detected magnetic resonance (ODMR) scheme, which is robust but still has some limitations. On the one hand, the simultaneously applied laser and microwave results in power broadening for the electronic spin resonance linewidth, which degrade the sensitivity of the NV magnetometer[38]. On the other hand, the heating effect of cw-microwave can increase the sample temperature by a few Kelvin[37], which is unfavorable for probing magnetic phenomena sensitively depending on the temperature[19,21,39].

In this work, we demonstrate cryogenic scanning magnetometry using a single NV center in a diamond probe. We show that by using a pulsed measurement scheme, the optimal magnetic field sensitivity that can be achieved with a commercial diamond probe is ~0.3 $\mu T Hz^{-\frac{1}{2}}$, and the microwave heating is

significantly reduced. With this setup, the magnetization of few-layer $CrBr_3$ samples is quantitatively studied, and the domain structure in a $CrBr_3$ bilayer is imaged in real space. We also study the magnetic domain evolution, and observe the domain wall pinning and reverse domain nucleation at defect sites.

## Results

**Cryogenic scanning NV magnetometry.** The schematics of the scanning NV magnetometry setup is depicted in Fig. 1a. The NV center is implanted in the apex of a pillar etched from a diamond cantilever, which is attached to the tuning fork of an atomic force microscope (AFM). We optically detect the NV spin state through the diamond cantilever supporting the pillar (see "Methods"). To efficiently apply microwave-driven field to the NV center, we deposit a coplanar waveguide on a $SiO_2/Si$ substrate and transfer the sample under test to one gap of it. In this work, the $CrBr_3$ samples are encapsulated with hexagonal boron nitride (hBN) on both sides, and transferred in a glovebox filled with pure nitrogen (see "Methods" for details of sample preparation). A heater and resistive thermometer is placed near the sample to control/monitor the sample temperature. The microscope head is suspended in an insertion tube filled with helium buffer gas, which is dipped in the liquid helium cryostat equipped with a set of vector superconducting coils. In the measurement,

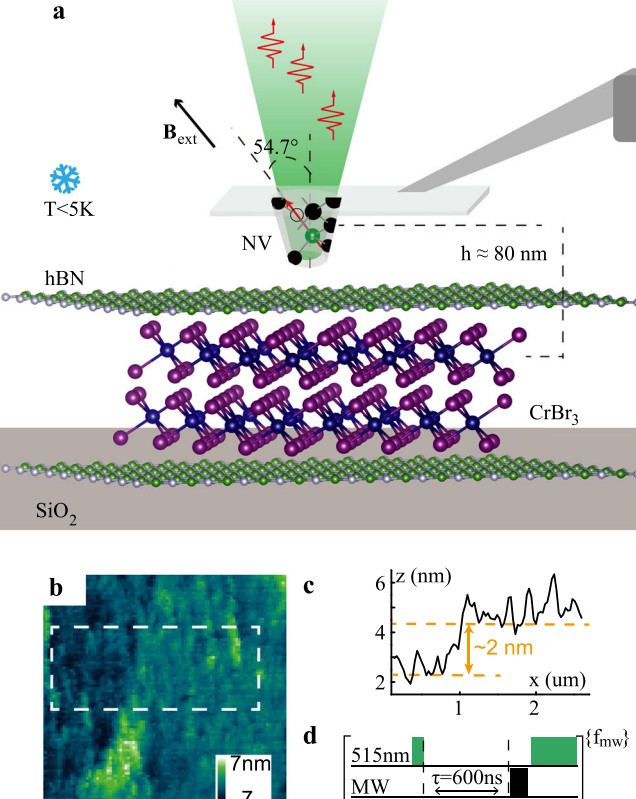

**Fig. 1 Cryogenic scanning magnetometry with a single NV center in the diamond tip. a** Schematic of the experiment. The stray magnetic field of the $CrBr_3$ bilayer is measured using a single NV center in a diamond probe attached to the tuning fork of the AFM. The system is placed in a liquid helium bath cryostat to maintain a temperature <5 K during the measurement. **b, c** A typical topography image of the measured area and the step at the edge of the sample, respectively. The scale bar is 1 μm. **d** Measurement sequence of the pulsed optically detected magnetic resonance (ODMR).

the AFM operates in a frequency modulation mode with a tip oscillation amplitude of ~1.5 nm. The topography of the sample is obtained from the AFM readout with a lateral spatial resolution of ~200 nm, which is limited by the diameter of the pillar apex. A typical in situ AFM image of part of the CrBr$_3$ sample is shown in Fig. 1b. Figure 1c shows the height of the step along the vertical direction by an average of the data in the dashed box in Fig. 1b. The 2 nm step height indicates a bilayer sample.

The stray magnetic field is mapped by measuring the electronic spin resonance spectrum via the pulsed ODMR scheme[38] at each pixel. The measurement sequence is shown in Fig. 1d. Instead of the cw-laser and -microwave, we apply short laser pulses and $\pi$ pulse microwave to the NV center. The ODMR curve is obtained by sweeping the microwave frequency, which is achieved by modulating a microwave signal with pairs of sinusoidal signals in quadrature via an IQ-mixer. All the control signals of the measurement sequences are generated using an arbitrary waveform generator so that the measurement sequence is well synchronized, and fast microwave frequency sweeping is realized by altering the frequency of the sinusoidal signals in each unit segment. In our experiment, we use commercial diamond probes (Qnami) with NV center formed from implanted $^{14}$N atom. Each spin transition from the $|m_s = 0\rangle$ to $|m_s = \pm 1\rangle$ states includes three sublevels, resulting from the hyperfine interaction with the nuclear spin of the $^{14}$N atom. The optimal sensitivity can be achieved by generating three pairs of sinusoidal signals to simultaneously excite the three hyperfine split transitions and detecting at the steepest slope of the ODMR resonance line. With a simple optimization, the sensitivity estimated from the ODMR curve is ~0.3 µTHz$^{-\frac{1}{2}}$ (see Supplementary Information for detail). This pulsed scheme also significantly decreases microwave heating compared with cw-ODMR[37]. In our experiment, the sample temperature only increases by a few hundred milli-Kelvin from the base temperature of ~4.2 K during measurements. Moreover, the magnetic field is measured via microwave pulses, which are applied 600 ns after the laser beam has been switched off. Thus, the measured stray magnetic field is not disturbed by laser-induced excitations, such as spin waves[40,41].

**Magnetization and magnetic domains of bilayer CrBr$_3$.** The stray magnetic field is obtained by fitting the ODMR curve with a Lorentzian lineshape. Figure 2a shows a typical stray magnetic field image of a CrBr$_3$ bilayer under a 2 mT external magnetic field after being cooled down under zero field. The external magnetic field is used to split the energy levels $|m_s = \pm 1\rangle$ so that the direction of the stray magnetic field can be determined. The external magnetic field direction is set parallel to the NV axis in all measurements in this work to avoid the mixing of spin states due to an off-axis magnetic field component[42]. The axis of all the NV centers in the (100)-oriented diamond cantilevers we used here is ~54.7° with respect to the vertical direction, as shown in Fig. 1a. The pixel size in this work is set to 30 nm and the data accumulation time is 2 s at each pixel. The resulting stray magnetic field map clearly shows magnetic domains with prominent positive and negative values of the magnetic field and domain walls with nearly zero fields. To reveal further details, we reconstruct the magnetization from the stray magnetic field using reverse-propagation protocol[34,37,43,44].

As discussed in more detail in the Supplementary Information, to uniquely determine the magnetization, we need some initial knowledge of the sample, such as the direction of spin polarization. It has been reported that few-layer CrBr$_3$ has an out-of-plane easy axis and can be polarized by a small external magnetic field of ~4 mT, while a relatively high external field ($B_\parallel^c \approx 0.44$ T) is required to polarize the spins in the in-plane

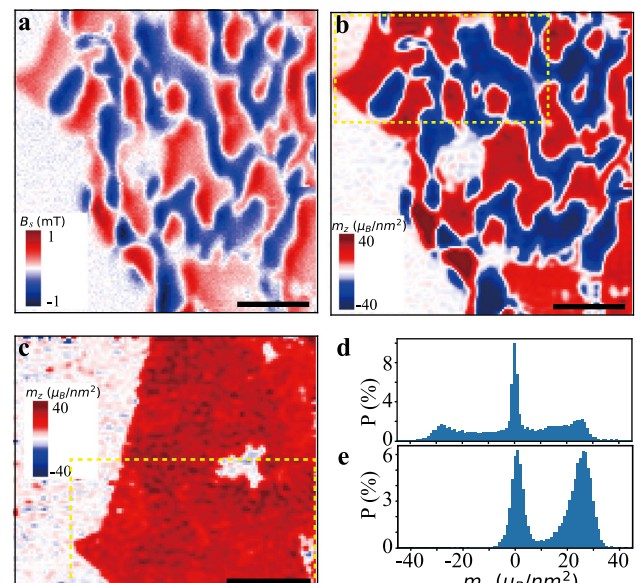

**Fig. 2 Magnetic domains and saturation magnetization. a** Stray magnetic field and **b** the reconstructed magnetization of a CrBr$_3$ bilayer under an external field of 2 mT along the NV axis. **c** Magnetization image at external magnetic field of 11 mT. The dashed boxes in **b** and **c** denote the common sample area in the two images. Scale bar is 1 µm for all images. **d**, **e** Histograms of the magnetization values in images **b** and **c**, respectively.

direction[45]. The in-plane external field component in this work is much lower than the critical field $B_\parallel^c$, allowing us to use the assumption of out-of-plane magnetization in the magnetization reconstruction. In addition, we neglect the finite thickness of the domain walls. Figure 2b presents the magnetization image reconstructed from the stray magnetic field image in Fig. 2a. It clearly shows the magnetic domain structure, with positive (negative) values, indicating the magnetization direction parallel (antiparallel) to the external magnetic field. The sample can be polarized by increasing the external magnetic field. Figure 2c shows a magnetization image taken at 11 mT external magnetic field. The common areas in Fig. 2b, c are marked with dashed boxes. The saturation magnetization can be estimated using the magnetization statistics of the two magnetization images, as shown by the histograms in Fig. 2d, e. Due to the sample imperfection, measurement error, and truncation error in the reconstruction, the reconstructed magnetization is distributed in a range around the zero-magnetization and the saturation magnetization values. The near-zero-magnetization pixels are mostly in domain walls, defects, and the non-sample area on the left part of the images. The saturation magnetization values are ~26(−28) and ~26 µ$_B$nm$^{-2}$, respectively, with µ$_B$ the Bohr magneton. These values are close to the 3 µ$_B$ saturation moment per Cr$^{3+}$ ion in CrBr$_3$ at 0 K, i.e., ~ 32 µ$_B$nm$^{-2}$ for a CrBr$_3$ bilayer[46].

**Domain wall pinning effect.** In addition to elucidating the magnetic domain structure of 2D magnets, our scanning magnetometry measurements enable a more detailed study of coercivity mechanisms in these systems. A multi-domain ferromagnet typically reverses its magnetization direction through processes, such as nucleation of reverse domains and their growth through domain wall motion[47,48]. Defects in the material alter the energy of the magnetic domain walls and hence affect domain wall motion. This behavior can be demonstrated by taking magnetization images, while varying the external magnetic field.

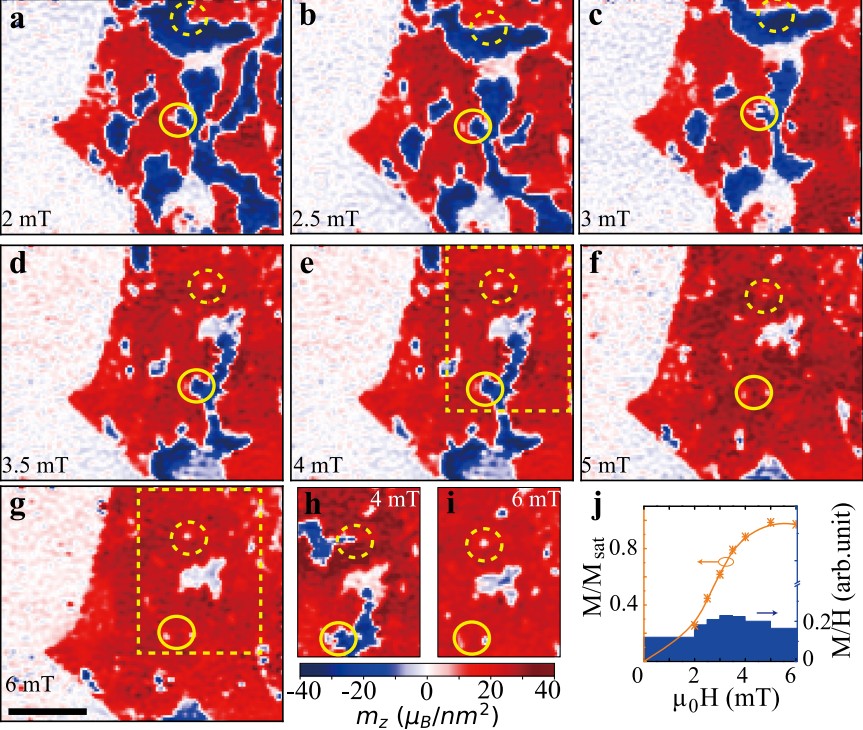

**Fig. 3 Magnetic domain evolution upon increasing the external magnetic field. a–g** Magnetization images taken successively at external magnetic fields of 2, 2.5, 3, 3.5, 4, 5, and 6 mT along the NV axis, respectively. The sample is thermally demagnetized by heating to 45 K and then cooling down under zero field. **h, i** Magnetization image of the sample area indicated by the dashed box in **e** and **g** during another thermal cycle at external magnetic fields of 4 and 6 mT, respectively. The solid and dashed yellow circles denote the positions of two representative pinning sites. See the Supplementary Information for the other magnetization images. The scale bar in **g** is 1 μm for all images. **j** Initial magnetization curves extracted from the magnetization images in **a–g**. The blue bars are the ratios of magnetization to external magnetic field.

Figure 3a–g shows magnetization images obtained while increasing the field from 2 to 6 mT after the sample is thermally demagnetized and cooled down under zero field. The area of positive (negative) domains grows (shrinks) with increasing field, as the domain walls move toward the negative domains. Before entirely disappearing, the negative domain size becomes very small, and only near-zero-magnetization spots of several tens of nanometer diameter are revealed in the magnetization images. As we are limited by the spatial resolution of the NV center (~80 nm in this work, determined by the distance between the NV center and the sample for the diamond probe), we cannot obtain the detailed magnetization pattern inside the spots. These spots are usually associated with defects, which alter the local switching field. Domain walls are pinned by these defects (see Fig. 3a–i, solid and dashed yellow circles). To confirm the pinning sites, we compare the magnetization images at 4 and 6 mT for different thermal cycles (dashed box in Fig. 3e, g and h, i). Though the magnetic domain structures are different upon successive thermal cycles, the positions of pinning sites are reproducible.

To verify that the pinning effect is a dominant coercivity mechanism, we extract the initial magnetization curve of the thermally demagnetized sample by estimating its average magnetization as $M/M_{sat} = \frac{N_+ - N_-}{N_+ + N_-}$, where $M_{sat}$ is the saturation magnetization and $N_+$ ($N_-$) is the number of pixels with evident positive (negative) magnetization (absolute value $>10\ \mu_B nm^{-2}$, according to the Fig. 2e). When pinning effects are negligible, the magnetic domain walls can move freely, resulting in a high initial magnetization even with a small external magnetic field. Defects, however, increase the energy barrier for the displacement of

domain walls. In samples with a large defect density, the magnetization increases slowly until the external magnetic field is large enough to overcome the pinning energy. The initial magnetization curve shown in Fig. 3j is measured at external magnetic fields >2 mT. We extend the curve to the origin using B-spline interpolation, assuming zero magnetization in a thermally demagnetized sample. The average permeability is very low when the field is <2 mT and it significantly increases when the field is >2 mT (see the blue bars in Fig. 3j), which is consistent with the behavior of a pinning effect dominated initial magnetization.

We use a similar approach to measure the magnetic hysteresis loop. Figure 4a shows the magnetization image at an external magnetic field of 2 mT after thermal demagnetization. We choose the area marked by the dashed box (#1) to analyse the magnetization, as we cycle the external magnetic field to saturate the magnetization in both the positive and negative z direction. Representative magnetization images are shown in Fig. 4b–i (see Supplementary Information for all images). Remarkably, a magnetic domain with the same irregular shape, but opposite magnetization direction is observed in Fig. 4e, i. The border of the magnetic domain (#2) is denoted by the dashed line. The hysteresis loop of the domain #2 is nearly rectangular, as shown by the red curve in Fig. 4j. There are several defects around the domain #2. Some of these defects act as nucleation centers of the reversed domains that appear immediately, when the external field is reduced from the saturation field. The domain walls move as the reversed domain grows and some of domain walls are finally pinned at the border of the magnetic domain (#2), which might

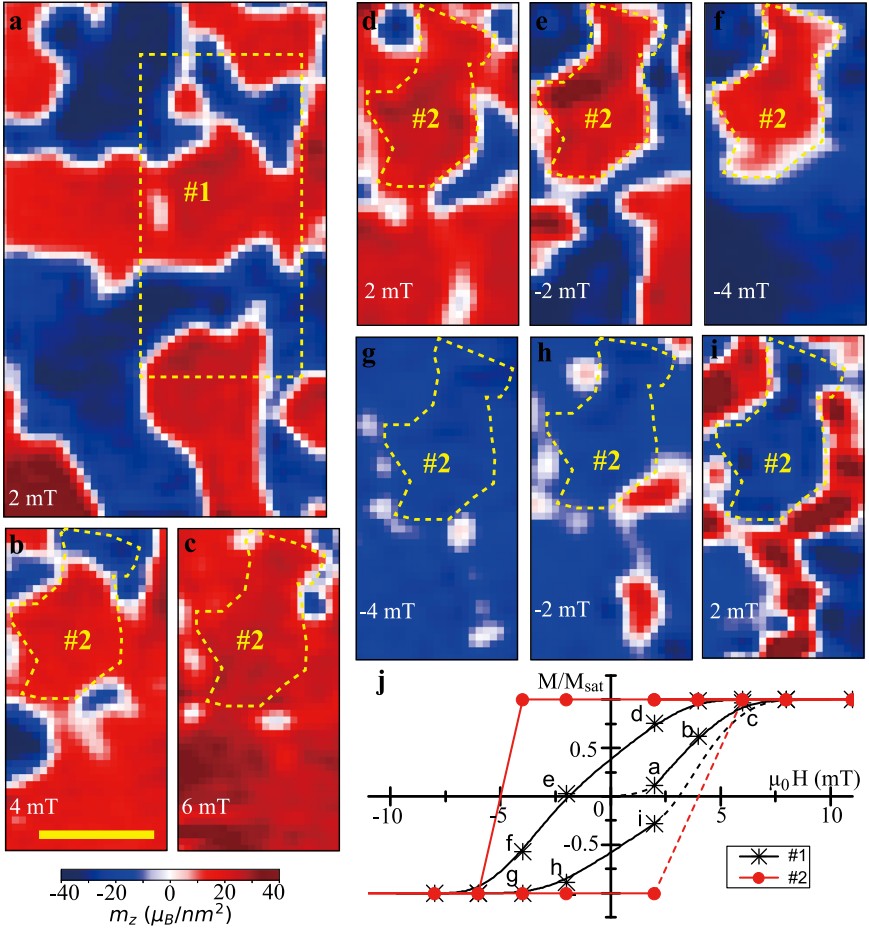

**Fig. 4 Magnetic hysteresis loop. a** Magnetic domains at 2 mT external magnetic field along the NV axis after being thermally demagnetized and cooled down under zero field. The dashed box denotes the area #1 used to analyze the hysteresis loop. **b–i** Representative magnetization images of area #1 on the hysteresis loop with the corresponding labels in **j**, and the dashed lines denote the area #2. The scale bar in **b** is 1 µm for all the images. **j** Hysteresis loop extracted from the magnetization images of areas #1 and #2, which are denoted by the black stars and red dots, respectively. The solid curve connects measured values, while the dashed curve is an extension to demagnetized and saturated states.

be a grain boundary. Therefore, the hysteresis loop averaged over area #1, which is represented by the black curve in Fig. 4j, indicates a lower coercive field.

## Discussion

To verify the observations, we reproduce the demagnetized ground state using a micromagnetic simulation with parameters within the ranges estimated from measurement of CrBr₃ bulk[49] (see "Methods" and Supplementary Information). This shows that the domain sizes and parameters and so on are all plausible. However, the micromagnetic simulation does not factor in the pinning effect, and therefore does not capture the reversal process itself. From this we can deduce that the magnetization (reversal) process is strongly dependent on the defects of the sample. In turn, this makes nanoscale magnetic imaging even more crucial, because the behavior cannot be predicted form simple simulation.

We also observe similar magnetic domain structures and domain wall pinning in other few-layer CrBr₃ samples, as presented in the Supplementary Information. The domain evolution measurement in a three-layer and four-layer sample indicates nearly rectangular hysteresis loops with higher coercive fields than domain #2 in the bilayer sample. Indeed, the influence of laser heating need to be determined as in previous works using NV center[31,50]. We show that the laser heating effect is negligible

in this work by measuring both the domain structure and magnetization at various laser power in a three-layer CrBr₃ sample.

In conclusion, we demonstrate a cryogenic scanning magnetometer based on a single NV center in a diamond probe with a pulsed measurement scheme. We show that our setup can achieve a high sensitivity and significantly reduce the microwave heating. Using this setup, we have studied magnetic domains in few-layer samples of CrBr₃ by quantitatively mapping the stray magnetic field. The magnetization of bilayer CrBr₃ is determined, and the magnetic domain evolution is observed in real space. We show that domain wall pinning is the dominant coercivity mechanism by observing the evolution of both the individual magnetic domains and the average magnetization, with changing external magnetic field. Our approach is also compatible with other pulsed measurement sequences that can be used to detect electronic spin resonance[51], nuclear magnetic resonance[32], and spin waves[52] in the 2D magnetic materials.

## Methods

**Sample fabrication**. The hBN flakes of 10–30 nm were mechanically exfoliated onto 90 nm SiO2/Si substrates, and examined by optical and atomic force microscopy under ambient conditions. Only atomically clean and smooth flakes were used for making samples. A V/Au (10/200 nm) microwave coplanar waveguide was deposited onto an 285 nm SiO2/Si substrate, using standard electron beam lithography with a bilayer resist (A4 495 and A4 950 polymethyl methacrylate) and electron beam evaporation. CrBr₃ crystals were exfoliated onto 90 nm SiO2/Si

substrates in an inert gas glovebox with water and oxygen concentration <0.1 p.p.m. The CrBr$_3$ flake thickness was identified by optical contrast and atomic force microscopy. The layer assembly was performed in the glovebox using a polymer-based dry transfer technique. The flakes were picked up sequentially: top hBN, CrBr$_3$, and bottom hBN. The resulting stacks were then transferred and released in a gap of the pre-patterned coplanar waveguide. In the resulting heterostructure, the CrBr$_3$ flake is fully encapsulated on both sides. Finally, the polymer was dissolved in chloroform for <5 min to minimize the exposure to ambient conditions.

**Confocal microscope**. The optics of the confocal microscope consists of the low-temperature objective (Attocube LT-APO/VISIR/0.82) with 0.82 numerical aperture and home-built optics head. The 515 nm excitation laser generated by an electrically driven laser diode is transmitted to the optics head through a polarization maintaining single-mode fiber and collimated by an objective lens. A pair of steering mirrors are used to align the beam for perpendicular incidence to the center of the objective. The NV center's fluorescence photons are also collected by the objective and transmitted through the same free-beam path to the optics head. The collected fluorescence photons are separated from the green laser beam via a dichroic mirror and then passed through a band-pass filter to further decrease background photons. Finally, the photons are coupled to a single-mode fiber and detected by a fiber-coupled single-photon detector.

**Stray magnetic field measurement**. With an external magnetic field applied along the NV axis, $B_{\parallel}$, the spin states $|m_s = 0\rangle$ and $|m_s = \pm 1\rangle$ exhibit Zeeman splitting of $f = D_s \pm \gamma_e B_{\parallel}$, where $D_s \approx 2.87$ GHz is the zero field splitting between levels $|m_s = \pm 1\rangle$ and $|m_s = 0\rangle$, and $\gamma_e = 28$ GHzT$^{-1}$ is the electronic spin gyromagnetic ratio. In the pulsed ODMR measurement, the NV center is optically initialized in spin state $|m_s = 0\rangle$. After a delay of $\tau = 600$ ns, a $\pi$-pulse (~80 ns) of microwave radiation is applied. If the microwave frequency is in resonance with one of the transitions, the NV center is driven to spin state $|m_s = \pm 1\rangle$. The population difference between the spin states $|m_s = 0\rangle$ and $|m_s = \pm 1\rangle$ can then be optically readout via fluorescence contrast. In our experiment, the laser pulse duration is 600 ns, and only the first 400 ns of the fluorescence photon signal is used in the data analysis. Details on the NV center ground-state energy levels and sensitivity can be found in Supplementary Information.

**Micromagnetic simulation**. To complement the experimental results, micromagnetic simulations of the systems magnetic ground state were conducted using MuMax3 (ref. [53]). Saturation magnetization was set to $M_s = 270$ kAm$^{-1}$, which is in accordance to the values measured here and reported in ref. [49]. Uniaxial magnetic anisotropy constant was assumed to be $K_u = 86$ kJm$^{-3}$ along the normal axis, which has been reported for bulk material[49]. A global exchange stiffness constant $A_{ex}$ was first roughly estimated from Curie temperature and experimentally observed domain wall width to lie within $10^{-12}$–$10^{-14}$ Jm$^{-1}$. Magnetization was initialized in a random configuration and then relaxed to the minimum energy state at zero external field. This was done for varying values of $A_{ex}$ from the interval estimated above.

The results for the normal magnetization component are shown in the Supplementary Information for a system trying to locally approximate the irregular shape of the real sample. This simulation assumed $A_{ex} = 3 \times 10^{-13}$ Jm$^{-1}$, which resulted in the closest match for the experiment. The simulation shows a domain structure that is qualitatively very similar to the experimental observations, which are further supported by this result. However, due to the limited capacity of the simulation to account for structural defects and pinning effects, the hysteresis behavior observed experimentally could not accurately be recreated by it. This further supports the conclusion that pinning is a major factor in this materials hysteresis.

## Data availability
The data that support the findings of this study are available from the corresponding author upon reasonable request

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

## Acknowledgements

The authors thank Dr. Patrick Maletinsky, Dr. Felipe Favaro de Oliveira, and Dr. Durga Dasari for the fruitful discussion, and Dr. Thomas Oeckinghaus and Dr. Roman Kolesov for the help with the experiment. J.W. acknowledges the Baden-Württemberg Foundation, the European Research Council (ERC; SMel grant agreement no. 742610). R.S. thanks the EU ASTERIQS. The work at U. Washington is mainly supported by DOE BES DE-SC0018171. Device fabrication is partially supported by AFOSR MURI program, grant no. FA9550-19-1-0390. The authors also acknowledge the use of the facilities and instrumentation supported by NSF MRSEC DMR-1719797.

## Author contributions

J.W. and X.X. supervised the project. Q.-C.S., A.B. and R.S. built the experimental setup. Q.-C.S. performed the experiment. T. Song and E.A. fabricated and characterized the samples. T. Shalomayeva provided experimental assistance. J.F. and J.G. performed the micromagnetic simulation. T.T. and K.W. provided and characterized bulk hBN crystals. Q.-C.S., T. Song, E.A., R.S., J.W. and X.X. wrote the paper with input from all authors. All authors discussed the results.

## Funding

## Competing interests

The authors declare no competing interests.
