## [Peer Review File · Nature Communications]

Reviewers' Comments:

Reviewer #1:

Remarks to the Author:

The paper by Sun et al reports a study of magnetic domains in a bilayer of CrBr₃ imaged by scanning NV magnetometry. It is certainly interesting to see images of the domain structure with such a high spatial resolution, while providing quantitative information (the absolute magnetization is measured) at the same time. Nevertheless, I don't think there is enough novelty to warrant publication in Nat Comm. Indeed, the methods are essentially identical to those in the 2019 Science paper that established the technique (Ref. 25). Here a different material is studied, CrBr₃ instead of CrI₃ in Ref. 25, but there is no fundamental difference otherwise. The present paper investigates more specifically the domain structure, but in terms of the methods there is no additional difficulty and in fact Ref. 25 also reported images of domains. So it comes down to the insights we gain from this study of CrBr₃, and I don't think these are particularly significant. Sure it is nice to see where the domain walls are pinned most strongly, but the overall behaviour is as expected from spatially-averaged hysteresis curves reported in previous studies of CrBr₃, modulo differences in sample quality due to local variations in defect density.

I also think the analysis is somewhat incomplete. Indeed, the authors start from a ZFC state and never apply more than 11 mT in their experiments. In SI Fig. 7, it is clear that 11 mT is not enough to fully magnetize the sample and small opposite domains remain (including probably some not visible because smaller than the spatial resolution). These small opposite domains then seed the growth of opposite domains when the field is reduced. Thus, the hysteresis curves reported in the main text are not exactly representative of a standard hysteresis measurement where a large field is applied to fully magnetize the sample on each side of the curve. It seems that the authors can apply up to 0.5 T in their setup, so it would be interesting to see how the domains evolve after applying 0.5 T then measuring at 11 mT, 8 mT etc. It is possible that in that case there would be no domain growth nucleating from inside the flake but only from the edges. In Fig. 4 I would also have liked to see the evolution of the magnetization near the edges of the flake, not just in the middle. That would tell us about the energy barrier at the edges, and would help to compare with previous measurements of CrBr₃ which typically encompassed sample edges due to limited spatial resolution.

Finally, the effect of laser heating should be quantified. The authors say that the laser increases the measured temperature by a few 100's of mK above the base 4.2 K, but this is measured by a remote sensor, and it is likely that the focused laser induces a temperature gradient, such that the temperature increase in the sample directly under the laser could be an order of magnitude more, as observed in [[dx.doi.org/10.1021/acs.nanolett.9b05071](https://doi.org/10.1021/acs.nanolett.9b05071)] in a comparable system. Of course measuring the local sample temperature is not trivial, but measuring magnetization vs incident laser power (as was done in Ref. 25) would at least give an indication that the local temperature may be closer to T_c than one would think. This is especially important if one wants to interpret the measured magnetization, which is found 20% lower than expected from bulk measurements (26 instead of 32 $\mu\text{B}/\text{nm}^2$).

Additional minor comments:

- I think the title should specify the material (CrBr₃) instead of being too general.
- In Fig. 3 and 4, scale bars seem to be missing

Reviewer #2:

Remarks to the Author:

The manuscript of Sun et al. presents study of magnetic domain structure and dynamics in atomically thin CrBr₃ at cryogenic temperatures using scanning NV center magnetometry. The presented magnetic imaging is of high quality and the data analysis is valid. The attained

quantitative reconstruction of the magnetization and of the magnetic domain structures is impressive.

The main question, however, is whether the manuscript meets the criteria for publication in Nature Communications in terms of importance of the results and their influence on the field.

The reported atomically thin CrBr₃ behaves as a regular thin ferromagnetic film. Its ferromagnetic behavior has been established already in Refs 26 and 27 and the saturation magnetization value has been determined in Ref 26. The authors present very nice high resolution imaging showing the domain formation which was not available previously for CrBr₃, but it reveals a conventional FM domain structure. The authors emphasize the importance of pinning of the domain walls by disorder. But this is the common source of hysteresis in FMs and the manuscript does not provide any insight into the nature of the underlying disorder or any quantitative information on the pinning parameters.

The authors also highlight the experimental technique. Scanning NV magnetometry is a well-established technique by now. Its operation at cryogenic temperatures is challenging, but has been already demonstrated in a number of papers including Refs 22, 23, 25. There is no statement of magnetic sensitivity and no comparison to other cryogenic magnetic imaging techniques including scanning nanoSQUIDs.

Reviewer #3:

Remarks to the Author:

The authors report measurement of domain configurations in the switching process of bilayer BrCr₃ using scanning NV center magnetometry. They were able to extract the value for saturation magnetization in BL BrCr₃, which is consistent with expectation. They further found that the domain pinning dominated the magnetic switching. Overall, I appreciate the data quality of the measurement, but I have some concerns about their claims and sample quality. I believe these need to be fully addressed before I can make a positive recommendation.

1. Regarding the sample quality, the coercive field observed in the bilayer sample in this work is around 4-5 mT. Since the field is applied along the NV axis (54.7 deg off z-axis), there needs to be a correction factor of $\cos(54.7\text{deg})=0.578$. Therefore, the z-component would be even smaller, ~2-3 mT. This coercive field seems to be much smaller compared to values reported in the literature (Refs 24, 26, 27) which is typically around 10 mT. Furthermore, in this work, the switching started before zero field. For example, in the M-H curve in Fig. 4, coming down from a full saturation, the M value drops to below 50% at zero field. While in earlier reports, the M-H curves all appear as somewhat square-ish, with little decrease at zero field. The authors mention that the rectangular loops in earlier reports might be due to the domain pinning. But how do authors justify that their observations are the common behaviors, instead of due to poor sample quality? What is the ordering temperature for this sample? In the supplemental materials, Fig 9 shows another thicker sample but the signal seems to disappear at 15 K. This is far below the reported T_c of 30 – 40 K. This also casts doubt about the sample quality.

2. I don't quite understand the argument on the top portion of page 8 about why the pinning effect is a dominant coercivity mechanism. The authors analyzed the evolution starting from a thermally demagnetized sample. The main argument is that at small field the permeability is small. But what can be considered as "small" can be arbitrary. Can authors provide some references on what it should look like for a process not dominated by pinning? What does the micromagnetic simulation say, (apparently, there is no pinning effect there)? Also Fig. 3j is quite misleading. The right-axis doesn't start from 0. Although it seems the M/H value is very low between 0 and 2 mT, it is in fact about 50% of the peak value.

3. Reference list needs to be fixed. Journal names are missing in many.

Reviewer#1 :

Comment 1:

The paper by Sun et al reports a study of magnetic domains in a bilayer of CrBr₃ imaged by scanning NV magnetometry. It is certainly interesting to see images of the domain structure with such a high spatial resolution, while providing quantitative information (the absolute magnetization is measured) at the same time. Nevertheless, I don't think there is enough novelty to warrant publication in Nat Comm.

Reply 1:

We thank the reviewer for his/her comments. Below is a point-to-point reply to the reviewer's comments.

Comment 2:

Indeed, the methods are essentially identical to those in the 2019 Science paper that established the technique (Ref. 25). Here a different material is studied, CrBr₃ instead of CrI₃ in Ref. 25, but there is no fundamental difference otherwise. The present paper investigates more specifically the domain structure, but in terms of the methods there is no additional difficulty and in fact Ref. 25 also reported images of domains.

Reply 2:

Below, we explain the difference in the detection method and the origin of the domains between our work and Ref. 25 are different.

First, we note that in ref.25 continuous wave (cw) optically detected magnetic resonance (ODMR) is used to measure the stray magnetic field. In this measurement scheme, cw microwave and laser are applied to the NV center and the fluorescence photons are also detected continuously. The cw ODMR is a robust method which is widely used in NV magnetometry. *However, simultaneously applied microwave and laser results in a power broadening of the NV ESR linewidth which degrades the sensitivity.* In addition, the cw microwave causes a strong heating effect, which is unfavorable for the study of magnetic phenomena sensitively depending on temperature.

To overcome these challenges, we employed a pulsed ODMR scheme, where the laser and microwave are pulsed and fluorescence photons are detected in a short time window. This is a technically challenging approach, because it needs great microwave performance, fast operation on the laser and microwave, and time resolved detection of the fluorescence. To implement this pulsed ODMR scheme, we designed impedance matched coplanar waveguides to apply microwave to the NV center. By mixing microwave signals with signals generated from an arbitrary waveform generator, we realized a high speed frequency sweeping (about 1 μ s per point), which can reduce the distortion in the ODMR spectrum caused by NV center's charge state switching and laser power instability. **With these improvements, we achieved a DC magnetic field sensitivity of about $0.3\mu T/\sqrt{Hz}$. This sensitivity is about one order of magnitude higher than previous reported scanning NV magnetometry (refs. 22, 23, and 25).**

Secondly, we want to point out that the domain structure observed in Ref.25 is mainly caused by the stacking order and layer-number differences in different regions, as explained by the authors on Page 3. Actually, in Ref.25 each structure domain of the CrI₃ has a single magnetic domain. Certainly, this is also interesting, but the origin of the domains is fundamentally different from the multi-domains we observed in a few-layer CrBr₃, as we explained in detail in the text.

We realized that the above points have not been clarified in our manuscript. We thank the reviewer for pointing it out. We significantly revised our manuscript and added additional data to make these points clearer.

Comment 3:

So it comes down to the insights we gain from this study of CrBr₃, and I don't think these are particularly significant. Sure it is nice to see where the domain walls are pinned most strongly, but the overall behavior is as expected from spatially-averaged hysteresis curves reported in previous studies of CrBr₃, modulo differences in sample quality due to local variations in defect density.

Reply 3:

As a magnetic insulator, CrBr₃ is one of the important building blocks of heterostructures. For example, recent work shows interesting phenomenon via magnetic proximity effect with CrBr₃, such as strong zero-field splitting of valley excitons in MoSe₂(PRL 124, 197401 (2020)), spin injection in Graphene (Advanced Materials, 2020, 32(16): 1908498), and 2D topological superconductivity (Nature 588, 424(2020)). Studying magnetic domain structures and evolution in CrBr₃ at nanoscale deepens the understanding of the material for designing and controlling the magnetization behavior of the device.

Although the multi-domain behavior can be inferred from the irregular hysteresis loop as in Ref.26 and Ref.27, it is difficult to obtain the domain size and domain structure limited by the spatial resolution. In addition, this indirect indication is not complete since Ref.26 shows the existent of multi-domains in few-layer CrBr₃ while in Ref.27, such signals have only been observed in thick CrBr₃ flakes. In this work, we unambiguously show the magnetic domains in few-layer CrBr₃ by nanoscale imaging.

Conventionally, the domain wall pinning is studied by measuring the initial magnetization curve not the hysteresis curve. The nucleation type magnet and pinning type magnet can have similar hysteresis curves while the initial magnetization curves are different, as shown in Fig 1.1 (Hysteresis in Magnetism: For Physicists, Materials Scientists, and Engineers 1998, Pages 347-390). To our knowledge, the initial magnetization curve of few-layer CrBr₃ has not been reported in previous work. In our work, we extracted the initial magnetization curve from the images taken at different external fields. In addition, thanks to the high spatial resolution, our technique also allows to observe the domain wall pinning and study the dynamics of individual domain wall in real space. The ability to locate the pinning sites allows further study of the nature of the defect by comparing the magnetic images with the crystal structure obtained by transmission electron microscopy technique

(Nano Lett. 20, 9, 6582–6589 (2020)).

Fig 1.1 Schematic representation of magnetization curves in nucleation-type and pinning-type magnets. (Hysteresis in Magnetism: For Physicists, Materials Scientists, and Engineers 1998, Pages 347-390)

Comment 4:

I also think the analysis is somewhat incomplete. Indeed, the authors start from a ZFC state and never apply more than 11 mT in their experiments. In SI Fig. 7, it is clear that 11 mT is not enough to fully magnetize the sample and small opposite domains remain (including probably some not visible because smaller than the spatial resolution). These small opposite domains then seed the growth of opposite domains when the field is reduced. Thus, the hysteresis curves reported in the main text are not exactly representative of a standard hysteresis measurement where a large field is applied to fully magnetize the sample on each side of the curve. It seems that the authors can apply up to 0.5 T in their setup, so it would be interesting to see how the domains evolve after applying 0.5 T then measuring at 11 mT, 8 mT etc. It is possible that in that case there would be no domain growth nucleating from inside the flake but only from the edges. In Fig. 4 I would also have liked to see the evolution of the magnetization near the edges of the flake, not just in the middle. That would tell us about the energy barrier at the edges, and would help to compare with previous measurements of CrBr₃ which typically encompassed sample edges due to limited spatial resolution.

Reply 4:

We thank the referee for pointing out the defect spots in the magnetic images. However, we don't think they are opposite domains. If there is any opposite domain (\downarrow) at the field of 11mT, then this domain acts as a seed of the reversed domain when the field is swept to -11mT. Therefore, a larger area around the domain is fully polarized and there is no opposite domain (\uparrow) at that spot, which can seed the reversed domain (\uparrow) when the field is swept back to 11mT. We find that these spots in SI Fig.7 either keep almost unchanged or seed the reversed domain (both \downarrow and \uparrow) when the field is swept in both directions. This behavior doesn't match that of the opposite domains we discussed above. Therefore, we think they are the defects spots of the sample rather than the opposite domains. Actually, the opposite domains usually have larger magnetization difference and larger size due to the

finite width domain walls. The magnetic signal should be stronger than the defect spots. The small nonzero magnetization of this spot is due to the imperfect magnetization reconstruction method, which needs to be improved in the future.

To study the domain reversal at the edge, we made a new sample to ensure high quality samples. We measured an area with a three-layer and four-layer CrBr_3 . The thermally demagnetized sample shows similar domain structures to the bilayer sample (see Fig. 1.2). We found that the three-layer CrBr_3 can be fully saturated at an external field of 11 mT and the coercive field is lower than 7 mT. This is consistent with our previous results that bilayer CrBr_3 saturates at 11 mT and the coercive of domain #2 in Fig.4 is about 5 mT. We studied the domain reversal after applying magnetic fields of 0.5T. The four-layer region has higher coercive field than the three-layer region. In the four-layer region, we observed the reversed domain first nucleated from the outer part of the sample and domain walls are pinned by the defects and grain boundary. Therefore, the inner part needs higher field to invert the magnetization direction.

We added the measurement results to the revised supplementary information.

Fig 1.2 Magnetic domain evolution upon varying the external magnetic field. a-h, Magnetization of a thermally demagnetized sample upon varying the external field. i-o, Measurements after the sample has been warmed up to 40 K and them cooled down to 4.2K at 0.5 T out-of-plane field.

Comment 5 :

Finally, the effect of laser heating should be quantified. The authors say that the laser increases the measured temperature by a few 100's of mK above the base 4.2 K, but this is measured by a remote sensor, and it is likely that the focused laser induces a temperature gradient, such that the temperature increase in the sample directly under the laser could be an order of magnitude more, as observed in [[dx.doi.org/10.1021/acs.nanolett.9b05071](https://doi.org/10.1021/acs.nanolett.9b05071)] in a comparable system. Of course measuring the local sample temperature is not trivial, but measuring magnetization vs incident laser power (as was done in Ref. 25) would at least give an indication that the local temperature may be closer to T_c than one would think. This is especially important if one wants to interpret the measured magnetization, which is found 20% lower than expected from bulk measurements (26 instead of 32 $\mu\text{B}/\text{nm}^2$).

Reply 5:

We thank the reviewer for the suggestion to measure the laser power dependent of magnetization. The temperature sensor is placed in a copper plate which has good thermal contact with the SiO_2/Si substrate. We didn't observe significant heating by the sensor when we only switch on the laser. The few 100 mK temperature incensement is mainly caused by the microwave.

We agree that the temperature at the laser spot is hard to characterize by the temperature sensor. To check the influence of laser, we measured the magnetization with different laser power as suggested by the reviewer (see Fig 1.3). The magnetization distributions obtained from the same area with different laser powers are shown in the figure below. The peaks on the left are contributed by the substrate and the those one the right are contributed by the sample. We found that the magnetization of the sample starts to decrease when the laser power is larger than $30\mu\text{W}$. All the images shown in the manuscript and the SI are obtained

with laser power of about $14\text{-}15\mu\text{W}$. We also measured the magnetic domain structure at different laser power and no obvious domain motion has been observed (Fig 1.4). So we can exclude the laser heating effect on the magnetization measurement. We note that a similar measurement performed in Ref.27 shows that the hysteresis loop of a CrBr_3 bilayer is invariant with laser power in a range of $20\text{-}100\mu\text{W}$.

Actually, the lower reconstructed magnetization should be attributed to the imperfection of the magnetization reconstruction algorithm. First, a constant distance between the NV center and the sample is assumed in the model. But this cannot be guaranteed due to the roughness of the substrate and the surface quality of the device. Second, in the reconstruction the stray field in region out of the measured area are treated as zero with an exponential decay region. So the truncation at the image boundary also reduces the reconstructed magnetization.

Fig 1.3 Laser power dependent of the magnetization of 3L CrBr₃

Fig 1.4 Magnetic domain structure at different laser power.

Comment 6:

I think the title should specify the material (CrBr₃) instead of being too general.

Reply 6:

We modified the title. Now it reads as, “Magnetic domains and domain wall pinning in atomically thin layered CrBr₃ revealed by nanoscale imaging”.

Comment 7:

In Fig. 3 and 4, scale bars seem to be missing

Reply 7:

In Fig.3 and Fig.4 all the images share the same scale bar, which are shown in Fig.3 (g) and Fig.4 (b). We made this point clear in the revised manuscript.

Reviewer#2:

Comment 1:

The manuscript of Sun et al. presents study of magnetic domain structure and dynamics in atomically thin CrBr₃ at cryogenic temperatures using scanning NV center magnetometry. The presented magnetic imaging is of high quality and the data analysis is valid. The attained quantitative reconstruction of the magnetization and of the magnetic domain structures is impressive. The main question, however, is whether the manuscript meets the criteria for publication in Nature Communications in terms of importance of the results and their influence on the field.

Reply 1:

We thank the reviewer for his/her comments. Below is a point-to-point reply to the reviewer's comments.

Comment 2:

The reported atomically thin CrBr₃ behaves as a regular thin ferromagnetic film. Its ferromagnetic behavior has been established already in Refs 26 and 27 and the saturation magnetization value has been determined in Ref 26. The authors present very nice high resolution imaging showing the domain formation which was not available previously for CrBr₃, but it reveals a conventional FM domain structure.

Reply 2:

We acknowledge that previous work has already determined a lot of the magnetic properties of CrBr₃, such as the ferromagnetic order, magnetization, and out-of-plane magnetic anisotropy. However, the multi-domain structure is determined by several energy terms and it is not a universal nature of 2D magnets. It has been shown that in another 2D ferromagnet Fe₃GeTe₂, the multi-domain structure only exists in thick flakes while thin flakes only show single domain (Nature Materials 17, 778–782(2018)). Although in previous works, the irregular hysteresis loops are interpreted as the result of multi-domains, it is difficult to extract the domain parameters from the spatial-average measurement. In addition, in Ref.26, the authors show the existent of multi-domains in few-layer CrBr₃ from the irregular shaped hysteresis loop. But in Ref.27, such signals have only been observed in thick CrBr₃ flakes. This conflict cannot be solved without imaging the magnetic domain structure at nanoscale.

Here, by using the high-spatial resolution scanning magnetometry, we unambiguously show the domain structure in bilayer CrBr₃. The small domain size and possible tuning mechanism by defect engineering make it a promising candidate for information storage. More importantly, as a 2D magnetic insulator, CrBr₃ is an important material to fabricate spintronic devices based on heterostructures. Recent works show interesting phenomenon via magnetic proximity effect with CrBr₃, such as strong zero-field splitting of valley excitons in MoSe₂ (PRL 124, 197401 (2020)), spin injection in Graphene (Advanced Materials, 2020, 32(16): 1908498), and 2D topological superconductivity (Nature 588, 424(2020)). The magnetic domain structure enriches the magnetization behavior of the heterostructure and could be employed in designing new devices.

Comment 3:

The authors emphasize the importance of pinning of the domain walls by disorder. But this is the common source of hysteresis in FMs and the manuscript does not provide any insight into the nature of the underlying disorder or any quantitative information on the pinning parameters.

Reply 3:

The hysteresis in FM is a result of interplay of different energy terms such as exchange energy, anisotropy energy, magnetostatic energy, Zeeman energy. The defects in the materials affect the hysteresis by locally modifying these energy terms, depending on the defect size, nature and density. But it is not a common source of hysteresis. For example, in the nucleation-type FM, the influence of defects on the hysteresis can be neglected. In our work, we for the first time show the bilayer CrBr₃ is a pinning-type FM. The initial magnetization indicates the statistic of the pinning energy. We also locate the position of pinning site, which was impossible to achieve in previous works. However, it is impossible to study nature of the pinning sites without knowing the crystal structure in that region. Actually, characterizing the atomic structure of encapsulated 2D magnets is another challenging work, due to the sample preparation and the hBN layer. Recently, a transmission electron microscopy technique with adhesion-enhanced grids has been shown to provide atomically scale resolution image of 2D CrBr₃(Nano Lett. 20, 9, 6582–6589 (2020)). It would be a promising way to study the nature of defect by combing these two techniques.

Comment 4:

The authors also highlight the experimental technique. Scanning NV magnetometry is a well-established technique by now. Its operation at cryogenic temperatures is challenging, but has been already demonstrated in a number of papers including Refs 22, 23, 25. There is no statement of magnetic sensitivity and no comparison to other cryogenic magnetic imaging techniques including scanning nanoSQUIDs.

Reply4:

We take this opportunity to discuss the improvement made in our work on the NV magnetometry in detail. Compared with Refs 22,23,25, we employ pulsed ODMR in the experiment instead of the cw ODMR. Our method can significantly reduce the power broadening of ODMR linewidth compared to previous work. **Therefore, we demonstrated a DC magnetic field sensitivity of about $0.3 \mu\text{T}/\sqrt{\text{Hz}}$, which is one order of magnitude improved compared with previous scanning NV magnetometry.** Moreover, this improvement also mitigates microwave heating effect which allows the study of magnetic phenomena sensitive to temperature. We added detail discussion of the sensitivity in the main text and supplementary information.

We added the comparison to other magnetic imaging techniques in the revised manuscript. The text reads as,

“Many high-spatial-resolution magnetic imaging techniques, such as magnetic force microscopy and Lorentz transmission electron microscopy, have been successfully used to

study the magnetic thin film materials. However, recent works show it remains challenging to study the atomically thin vdW magnets with them due to the weak signal level^{17,20}. Scanning superconducting quantum interference device (SQUID) can achieve high magnetic field sensitivity even with a probe diameter of about 50 nm^{26,27}. It works well at temperatures below a few Kelvin, but the work temperature range is limited by the low temperature superconducting material used to fabricate probe. In contrast, the negatively charged nitrogen-vacancy (NV) center has been demonstrated as a high sensitivity magnetometer with operational temperatures from below one to several hundreds of Kelvin²⁸, which is suitable to probe most of the discovered vdW magnets.”

Reviewer#3:

Comment 1:

The authors report measurement of domain configurations in the switching process of bilayer BrCr₃ using scanning NV center magnetometry. They were able to extract the value for saturation magnetization in BL BrCr₃, which is consistent with expectation. They further found that the domain pinning dominated the magnetic switching. Overall, I appreciate the data quality of the measurement, but I have some concerns about their claims and sample quality. I believe these need to be fully addressed before I can make a positive recommendation.

Reply 1:

We thank the referee for his/her comments. Below is a point-to-point reply to the referee's comments.

Comment 2:

Regarding the sample quality, the coercive field observed in the bilayer sample in this work is around 4-5 mT. Since the field is applied along the NV axis (54.7 deg off z-axis), there needs to be a correction factor of $\cos(54.7\text{deg})=0.578$. Therefore, the z-component would be even smaller, ~2-3 mT. This coercive field seems to be much smaller compared to values reported in the literature (Refs 24, 26, 27) which is typically around 10 mT. Furthermore, in this work, the switching started before zero field. For example, in the M-H curve in Fig. 4, coming down from a full saturation, the M value drops to below 50% at zero field. While in earlier reports, the M-H curves all appear as somewhat square shaped, with little decrease at zero field. The authors mention that the rectangular loops in earlier reports might be due to the domain pinning. But how do authors justify that their observations are the common behaviors, instead of due to poor sample quality?

Reply2:

Actually, the hysteresis loop with different field orientation has been measured in Ref.26. From Fig 2.2 in Ref. 26, one can find that the angle dependent of the magnetization is not significant when the angle to the vertical direction is less than 60°. Therefore, in our work, the coercive field before correction should be close to the coercive field when an out-of-plane field is applied.

All the samples we measured are exfoliated from CrBr₃ bulk (HQ graphene) with the procedure explained in the Methods of the manuscript. The encapsulated samples are only exposed in the air for about 10 minutes when they are mounted to our setup to avoid sample degradation. We agree that the M-H curve in Fig.4 is not a common behavior, which might be strongly modified by the local defects. In this revision we also added new measurement results from a 3L and 4L CrBr₃, which shows higher coercive field than the bilayer sample. But the coercive field are still lower than 10mT. Actually, the coercive field of bilayer CrBr₃ extracted from the hysteresis loop in Ref.27 (Fig.4, Fig.S4, and Fig.S6) is also about 5 mT. We think the difference coercive fields measured in previous work is another interesting topic for further study.

We modified the discussion in the manuscript. The text reads as,

“Remarkably, a magnetic domain with the same irregular shape but opposite magnetization direction is observed in Figs.4 (e) and (i). The border of the magnetic domain (#2) is denoted by the dashed line. The hysteresis loop of the domain #2 is nearly rectangular, as shown by the red curve in Fig.4 (j). There are several defects around the domain #2. Some of these defects act as nucleation centers of the reversed domains which appear immediately when the external field is reduced from the saturation field. The domain walls move as the reversed domain growing and some of domain walls are finally pinned at the border of the magnetic domain (#2), which might be a grain boundary. Therefore, the hysteresis loop averaged over area #1, which is represented by the black curve in Fig.4 (j), indicates a lower coercive field.”

Fig 3.1 Angular dependence of the amplitude of the magnetic hysteresis ΔM as a function of H and θ . (ref. 26)

Comment 3:

What is the ordering temperature for this sample? In the supplemental materials, Fig 9 shows another thicker sample but the signal seems to disappear at 15 K. This is far below the reported T_c of 30 – 40 K. This also casts doubt about the sample quality.

Reply 3:

The Fig.9 in supplementary information, shows the disappearing of domain structure in the iso-field image. This might be limited by the sensitivity of the NV center. Note that there is still a weak stray field signal from the sample. So it is not the T_c of the sample. We didn't get enough images with that sample before the degradation of the NV center in the measurement. We will remove this images to avoid any misunderstanding. Instead, we will add the measurement results of a 3L sample, which shows that the T_c is above 30K.

Fig 3.2 stray field of a 3L sample measured at 28 K (left) and 30 K (right).

Comment 3:

I don't quite understand the argument on the top portion of page 8 about why the pinning effect is a dominant coercivity mechanism. The authors analyzed the evolution starting from a thermally demagnetized sample. The main argument is that at small field the permeability is small. But what can be considered as "small" can be arbitrary. Can authors provide some references on what it should look like for a process not dominated by pinning? What does the micromagnetic simulation say, (apparently, there is no pinning effect there)? Also Fig. 3j is quite misleading. The right-axis doesn't start from 0. Although it seems the M/H value is very low between 0 and 2 mT, it is in fact about 50% of the peak value.

Reply 3:

The nucleation type and pinning type magnet is a qualitative concept. Typically, it can be determined by comparing the initial magnetization curve with the saturation hysteresis loop. As shown in the figure below, for the pinning type magnet, the initial magnetization curve saturates at field values on the same order of the coercive field, while for the nucleation type magnet, saturation happens at much lower field than the coercive field. (Hysteresis in Magnetism: For Physicists, Materials Scientists, and Engineers 1998, Pages 347-390).

The micromagnetic simulation can reproduce the demagnetized ground state. This shows that the domain sizes and parameters and so on are all plausible. However, the micromagnetic simulation does explicitly not capture the reversal process itself. From this we can deduce that the magnetization (reversal) process is strongly dependent on the defects of the sample. In turn, this makes the nano-resolved experiment even more crucial, because the behavior cannot be predicted from simple simulation.

Fig.3 j has been modified in the revised manuscript. We thank the reviewer for pointing out this issue.

Comment 4:

Reference list needs to be fixed. Journal names are missing in many.

Reply 4:

We thank the reviewer for pointing out the missing journal names in references. We fixed this problem in this revision.

Fig 3.3 Schematic representation of magnetization curves in nucleation-type and pinning-type magnets. (Hysteresis in Magnetism: For Physicists, Materials Scientists, and Engineers 1998, Pages 347-390)

Reviewers' Comments:

Reviewer #1:

Remarks to the Author:

The authors presented new/expanded arguments and data in response to my concerns about novelty and analysis. The new data is very welcome but my assessment of the novelty of the technique remains unchanged (see details below), and so overall I don't think the paper is suitable for Nat Comm in its present form.

The authors now argue that a key novelty of their work is the use of pulsed ODMR, which they say is technically challenging and improves the sensitivity compared to CW ODMR used e.g. in Ref. 25. They even changed the abstract to make this point front and centre. I think it is a minor point that does not deserve such an emphasis, mainly because the quality of the data looks very similar to that in Ref. 25, i.e. similar SNR for similar field amplitudes in the 100 uT range, which means the improvement in sensitivity (if genuine) does not provide a clearly visible benefit. I'm also cautious with claims about sensitivity enhancements in this context because it depends on definition and conditions (e.g. laser intensity, NV spin T_2^* , etc). Finally, pulsed ODMR is not particularly challenging to implement, and in fact I would assume that most previous works which used CW ODMR did so not for simplicity but because it was a better choice overall given their conditions (especially in the low laser intensity regime desired). At the end of the day, what matters is the pixel-to-pixel noise in the magnetic field images for a given per-pixel acquisition time and for a given level of "invasiveness" (i.e. how much laser/microwave heating is generated to reach this noise), and it seems to me that the present work is quite comparable to Ref. 25 in that regard. In summary, a discussion of the technical improvements made in this work (like the fast frequency sweeping) is certainly useful, but I don't think focusing on the protocol (pulsed vs CW) and making it a central argument is justified.

Instead, I think the emphasis of the paper should be on the new insights gained on the magnetic domain structure in ultrathin CrBr₃, rather than on the technique. In particular, the abstract mentions the pulsed ODMR protocol and the sensitivity and finishes with "Our work highlights scanning nitrogen-vacancy center magnetometry as...", but says nothing about the actual results and their significance. Given the impressive results and the quality of the data, I think a properly revised text could be suitable for Nat Comm.

Reviewer #2:

Remarks to the Author:

The authors have improved their manuscript substantially. It is now acceptable for publication.

Reviewer #3:

Remarks to the Author:

I couldn't seem to tell if Fig. 3j was modified. Otherwise, my concerns have been addressed and I am ok with recommending for publication.

Responses to reviewer comments

Reviewer #1 (Remarks to the Author):

Comment 0:

The authors presented new/expanded arguments and data in response to my concerns about novelty and analysis. The new data is very welcome but my assessment of the novelty of the technique remains unchanged (see details below), and so overall I don't think the paper is suitable for Nat Comm in its present form.

Reply 0:

We thank the reviewer for acknowledging our new data. Below is a point-to-point reply to the reviewer's comments.

Comment 1:

The authors now argue that a key novelty of their work is the use of pulsed ODMR, which they say is technically challenging and improves the sensitivity compared to CW ODMR used e.g. in Ref. 25. They even changed the abstract to make this point front and centre. I think it is a minor point that does not deserve such an emphasis, mainly because the quality of the data looks very similar to that in Ref. 25, i.e. similar SNR for similar field amplitudes in the 100 uT range, which means the improvement in sensitivity (if genuine) does not provide a clearly visible benefit. I'm also cautious with claims about sensitivity enhancements in this context because it depends on definition and conditions (e.g. laser intensity, NV spin T2*, etc). Finally, pulsed ODMR is not particularly challenging to implement, and in fact I would assume that most previous works which used CW ODMR did so not for simplicity but because it was a better choice overall given their conditions (especially in the low laser intensity regime desired). At the end of the day, what matters is the pixel-to-pixel noise in the magnetic field images for a given per-pixel acquisition time and for a given level of "invasiveness" (i.e. how much laser/microwave heating is generated to reach this noise), and it seems to me that the present work is quite comparable to Ref. 25 in that regard. In summary, a discussion of the technical improvements made in this work (like the fast frequency sweeping) is certainly useful, but I don't think focusing on the protocol (pulsed vs CW) and making it a central argument is justified.

Reply 1:

We thank the reviewer for raising a valuable discussion on the selection of detection methods. We agree that there are many factors that may affect the sensitivity. In response to the reviewer comments we removed the words "*This sensitivity is about one order of magnitude higher than previous works*" in the main text.

Nevertheless, we would like to comment on the importance to use pulsed ODMR in the present work. Indeed, the magnetic imaging technique used to study materials should be chosen properly by taking the measurement requirements and technical limitations into consideration, such as the signal level of the sample (sensitivity), temperature to stabilize the phase (heating), and size of the magnetic structures (spatial resolution). As we discussed in the supplementary information, pulsed ODMR can be conducted by driving the NV transition with either a large Rabi frequency or a low Rabi frequency. The latter method offers a higher sensitivity than the former one, while the former one offers much larger magnetic field measurement range. Importantly, for CrBr₃, the stray field near the domain walls is much larger than the minimal detectable field $\delta B = \eta/\sqrt{\tau}$, which is determined by the optimized sensitivity η and measurement time in our measurement $\tau = 2$ s. Therefore, we use pulsed ODMR with a large Rabi frequency to measure the magnetic domain structures which are shown in the main text. That's why the SNR is similar to the images in Ref. 25. We clarified this issue in the revised manuscript. We agree that it is possible that previous works use CW ODMR due to some reasons other than technical challenging. However, we don't think low laser power is a crucial issue in most of the

previous works because laser power used in these works is larger than that used in our work (15 μW). For example, in Ref.25, the laser power is 40 μW .

On the other hand, the highest sensitivity is realized using pulsed ODMR with low Rabi frequency. Actually, the sensitivity reported in previous works and our work are consistent with that reported in a systematically study of the sensitivity enhancements of NV centers in CVD diamond (PRB 84 (19), 2011), which shows the sensitivity obtained in an optimized CW-ODMR and pulse ODMR is $2 \mu\text{T}/\sqrt{\text{Hz}}$ and $0.3 \mu\text{T}/\sqrt{\text{Hz}}$, respectively.

Comment 2:

Instead, I think the emphasis of the paper should be on the new insights gained on the magnetic domain structure in ultrathin CrBr₃, rather than on the technique. In particular, the abstract mentions the pulsed ODMR protocol and the sensitivity and finishes with “Our work highlights scanning nitrogen-vacancy center magnetometry as...”, but says nothing about the actual results and their significance. Given the impressive results and the quality of the data, I think a properly revised text could be suitable for Nat Comm.

Reply 2:

We thank the reviewer for the suggestion. We rewrote the abstract. Now it reads as,

“The emergence of atomically thin van der Waals magnets provides a new platform for the studies of two-dimensional magnetism and its applications. However, the widely used measurement methods in recent studies cannot provide quantitative information of the magnetization nor achieve nanoscale spatial resolution. These capabilities are essential to explore the rich properties of magnetic domains and spin textures. Here, we employ cryogenic scanning magnetometry using a single-electron spin of a nitrogen-vacancy center in a diamond probe to unambiguously prove the existence of magnetic domains and study their dynamics in atomically thin CrBr₃. By controlling the magnetic domain evolution as a function of magnetic field, we find that the pinning effect is a dominant coercivity mechanism and determine the magnetization of a CrBr₃ bilayer to be about $26 \mu_B/\text{nm}^2$. The high spatial resolution of this technique enables imaging of magnetic domains and allows to locate the sites of defects that pin the domain walls and nucleate the reverse domains. Our work highlights scanning nitrogen-vacancy center magnetometry as a quantitative probe to explore nanoscale features in two-dimensional magnets.”

Reviewer #2 (Remarks to the Author):

Comment 0:

The authors have improved their manuscript substantially. It is now acceptable for publication.

Reply 0:

We thank the reviewer for acknowledging our revision. We are glad to see that the reviewer supports the publication of our paper.

Reviewer #3 (Remarks to the Author):

Comment 0:

I couldn't seem to tell if Fig. 3j was modified. Otherwise, my concerns have been addressed and I am ok with recommending for publication.

Reply 0:

We thank the reviewer for pointing out this oversight. We included the correct figure in this revision. We thank the reviewers for his/her time and effort to review our manuscript.

Reviewers' Comments:

Reviewer #1:

Remarks to the Author:

The authors have addressed my concerns and the revised abstract now presents a complete picture of the relevance of the results to the 2D materials community. I recommend publication in Nat Comm, and congratulate the authors on the beautiful data and thorough investigation.

Response to Reviewers' comments

Reviewer #1 (Remarks to the Author):

Comment 0: The authors have addressed my concerns and the revised abstract now presents a complete picture of the relevance of the results to the 2D materials community. I recommend publication in Nat Comm, and congratulate the authors on the beautiful data and thorough investigation.

Reply 0: We thank the reviewer for his/her time and effort to review our manuscript. We are glad to see that the reviewer now recommends the publication of our manuscript in Nature Common.